# Perinatal hormones favor CC17 group B *Streptococcus* intestinal translocation through M cells and hypervirulence in neonates

Constantin Hays[1,2,3,4], Gérald Touak[1,2,3], Abdelouhab Bouaboud[1,2,3], Agnès Fouet[1,2,3], Julie Guignot[1,2,3], Claire Poyart[1,2,3,4,5], Asmaa Tazi[1,2,3,4,5]*

[1]Institut Cochin, Team Bacteria and Perinatality, INSERM U1016, Paris, France; [2]CNRS UMR 8104, Paris, France; [3]Paris Descartes University, Paris, France; [4]Department of Bacteriology, University Hospitals Paris Centre-Cochin, Assistance Publique – Hôpitaux de Paris, Paris, France; [5]National Reference Center for Streptococci, Paris, France

**Abstract** Group B *Streptococcus* (GBS) is the leading cause of invasive bacterial neonatal infections. Late-onset diseases (LOD) occur between 7 and 89 days of life and are largely due to the CC17 GBS hypervirulent clone. We studied the impact of estradiol (E2) and progesterone (P4), which impregnate the fetus during pregnancy, on GBS neonatal infection in cellular and mouse models of hormonal exposure corresponding to concentrations found at birth (E2-P4 $C_0$) and over 7 days old (E2-P4 $C_7$). Using representative GBS isolates, we show that E2-P4 $C_7$ concentrations specifically favor CC17 GBS meningitis following mice oral infection. CC17 GBS crosses the intestinal barrier through M cells. This process mediated by the CC17-specific surface protein Srr2 is enhanced by E2-P4 $C_7$ concentrations which promote M cell differentiation and CC17 GBS invasiveness. Our findings provide an explanation for CC17 GBS responsibility in LOD in link with neonatal gastrointestinal tract maturation and hormonal imprint.

DOI: https://doi.org/10.7554/eLife.48772.001

*For correspondence:
asmaa.tazi@aphp.fr

**Competing interests:** The authors declare that no competing interests exist.

## Introduction

Group B *Streptococcus* (*Streptococcus agalactiae*, GBS) is a natural inhabitant of the gastrointestinal and vaginal flora of 10% to 30% healthy individuals. This Gram positive encapsulated bacterium can also turn into a deadly pathogen in neonates and is recognized as a leading cause of neonatal invasive infections (bacteremia, meningitis) (*Edwards et al., 2011*). Despite appropriate antibiotic therapy, the global burden of GBS neonatal infections remains substantial, with up to 10% mortality and 30% neurologic sequelae in surviving infants (*Libster et al., 2012*). Two GBS-associated syndromes are distinguished in neonates, the early-onset disease (EOD) which occurs within 48 hr after birth in 90% of the cases, and the late-onset disease (LOD) which occurs between 7 and 89 days of life (*Edwards et al., 2011*). EOD primarily manifests as a pneumonia and a bacteremia subsequent to the inhalation by the neonate of GBS contaminated amniotic fluid or vaginal secretions during delivery (*Edwards et al., 2011*). In contrast, the mode of transmission and the infection route of LOD, characterized by a bacteremia without focus and a high rate of associated meningitis, remain elusive, and although the gastrointestinal tract represents the most likely portal of entry, the circumstances leading to LOD remain largely unknown (*Tazi et al., 2010*).

Worldwide epidemiological data have identified that GBS strains are distributed among five major clonal complexes (CC) (*Da Cunha et al., 2014*). One particular GBS clone, almost exclusively

of capsular serotype III and designated the hypervirulent CC17 clone, is strongly associated with LOD (60% to 80% of the cases) and with meningitis (approx. 80% of the cases) (*Bekker et al., 2014*; *Jones et al., 2006*; *Joubrel et al., 2015*; *Seale et al., 2016*). CC17 GBS expresses specific surface proteins which promote intestinal colonization and confer blood brain barrier (BBB) translocation abilities in animal models (*Six et al., 2015*; *Tazi et al., 2010*). Remarkably, this clone is less responsible for EOD and largely underrepresented in GBS adult diseases (*Huber et al., 2011*; *Martins et al., 2012*) implying that specific host factors might be involved in CC17 GBS invasiveness in >7 day-old neonates.

One particular aspect of the perinatal life is the very high concentration of pregnancy-related hormones, especially estradiol (E2) and progesterone (P4), to which neonates are exposed all along pregnancy. This hormonal exposure leads to neonatal circulating concentrations at birth which are more than 500 times superior to those of a male adult (*Lagiou et al., 2014*). These levels drop during the first 3 days of life to stabilize for several months at concentrations still 5 to 50 times higher than those of an adult (*Khashana et al., 2016*; *Kuijper et al., 2013*). Interestingly, sex hormones modulate the immune response and the permeability of cellular barriers in a dose-dependent manner (*Hata et al., 2014*; *Kovats, 2015*; *Robinson and Klein, 2012*; *Straub, 2007*). Thus, E2 and P4 concentrations found at a given time-period of the neonate's development could promote GBS and more particularly CC17 GBS invasiveness and LOD.

Using an approach combining murine and cellular models of infection with one CC17 and one non-CC17 representative GBS isolates, both of capsular serotype III and responsible for neonatal disease, we demonstrate that hormonal exposure to E2 and P4 concentrations found in 7-day-old neonates (E2-P4 $C_7$) specifically favors CC17 GBS pathogeny in comparison to E2 and P4 levels found at birth (E2-P4 $C_0$). The increased invasiveness of CC17 GBS in E2-P4 $C_7$ condition is linked to an enhanced crossing of the intestinal barrier through M cells which is mediated by the CC17-specific surface protein Srr2.

## Results

### E2-P4 circulating levels modulate the severity of CC17 GBS meningitis in mice

We used a murine model of meningitis following oral gavage and intestinal translocation to determine whether hormonal concentrations could impact GBS and more particularly CC17 GBS infection. Prior to infection, specific pathogen-free (SPF) mice were submitted to a hormonal impregnation (*Cai et al., 2014*; *Hata et al., 2014*) leading to circulating E2 and P4 levels measured 4 hr after the last hormonal administration equivalent to those found at birth ($C_0$ mice) or 7 days later ($C_7$ mice), that is approx. $1.10^{-8}$ and $1.10^{-6}$ M at birth, and $5.10^{-10}$ and $5.10^{-8}$ M at day 7, for E2 and P4, respectively (*Figure 1a*). Control mice were administered the vehicle alone and showed E2 and P4 levels approx. 100 times and 1000 times lower than those of $C_7$ and $C_0$ mice, respectively. Four hours after the last hormonal injection, mice were gavaged either by a CC17 GBS (strain BM110, capsular serotype III) or by a non-CC17 GBS (strain NEM316, capsular serotype III) belonging to the clonal complex 23 (CC23 GBS). Indeed, CC23 GBS are representative of capsular serotype III non-CC17 GBS isolates responsible for neonatal disease (*Da Cunha et al., 2014*). Mice subsequent gavage by $2.10^{10}$ colony forming units (CFU) of CC17 GBS led to higher bacterial counts (~10 fold) in the brain of $C_7$ mice, 2 and 24 hr after infection, compared to $C_0$ and control mice (*Figure 1b*), demonstrating that E2-P4 $C_7$ concentrations contribute to the severity of CC17 GBS infection. In contrast, the same experiment using CC23 GBS did not show any significant difference between the 3 groups of mice, and lower CFU counts in the brain of CC23 compared to CC17-infected $C_7$ mice (*Figure 1b*) indicating that E2-P4 $C_7$ concentrations specifically favor CC17 GBS infection in our experimental model. CC17 GBS and CC23 GBS growth *in vitro* was not altered by E2-P4 concentrations (*Figure 1—figure supplement 1*), suggesting that these results were not due a direct impact of E2 and P4 on bacterial multiplication in mice.

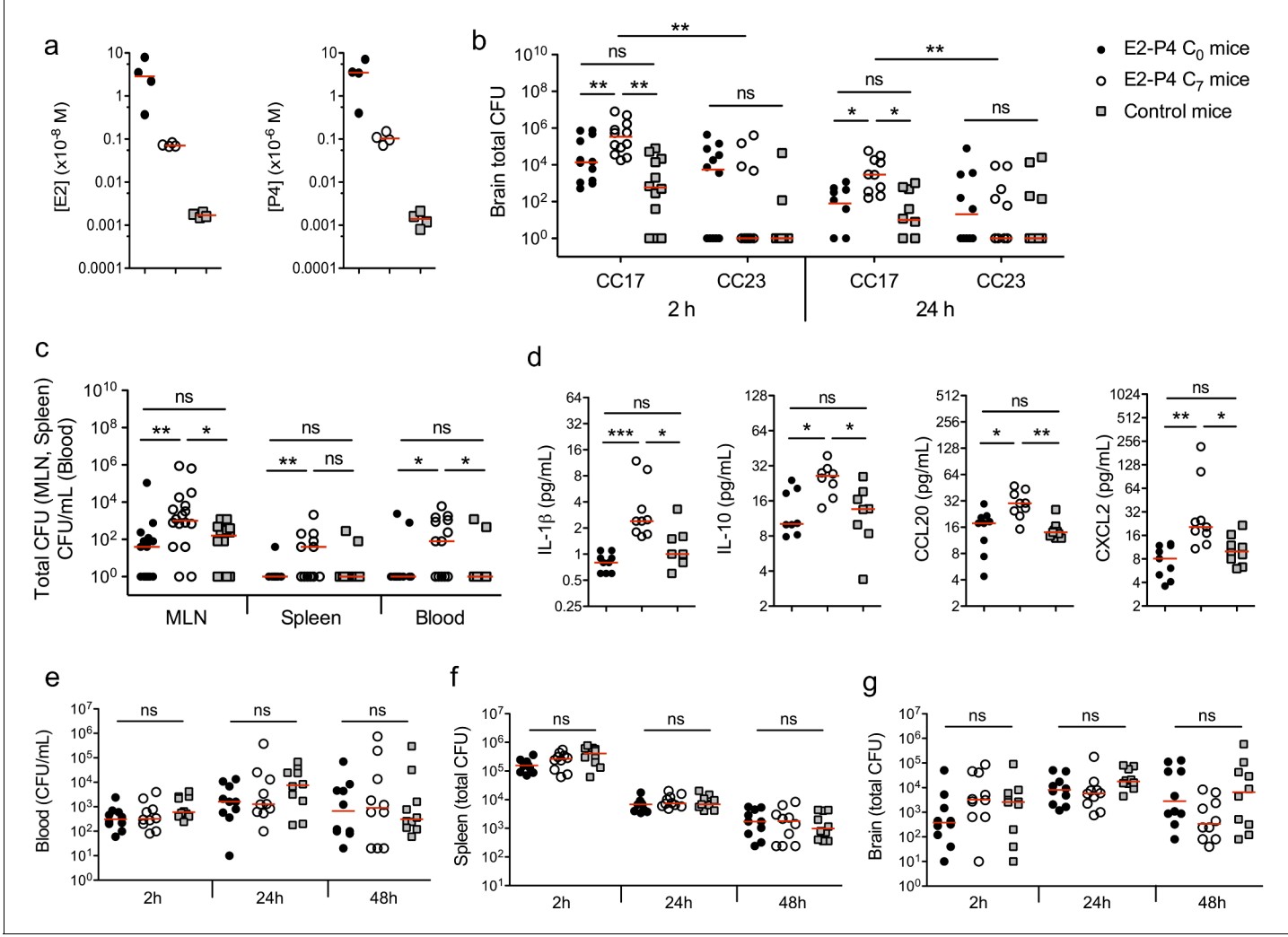

**Figure 1.** E2-P4 hormone levels modulate CC17 GBS dissemination and the severity of meningitis following oral infection in mice. SPF 3-week-old mice were administered E2-P4 cocktails subcutaneously for four consecutive days leading to E2-P4 circulating levels equivalent to those found in neonates at birth (E2-P4 $C_0$ mice) or 7 days later (E2-P4 $C_7$ mice). Control mice were administered vehicle alone. (a) Serum levels of E2 and P4 in the 3 groups of mice measured 4 hr after the last hormonal administration (n = 4 mice per group). (b to d) Mice were gavaged with representative CC17 (strain BM110) or CC23 (strain NEM316) GBS isolates ($2.10^{10}$ CFU). (b) Total CFU counts in the brain 2 hr (n = 12 mice per group) and 24 hr (n = 10 mice per group) after infection by CC17 and CC23 GBS. (c) Total CFU counts in the mesenteric lymph nodes (MLN, n = 16 mice per group), spleen (n = 12 mice per group), and blood circulating bacteria in CFU/mL (n = 12 mice per group) 2 hr after infection by CC17 GBS. (b, c) $10^0$ represents the detection threshold. (d) Serum levels of the cytokines IL-1β, IL-10, CCL20 and CXCL2 2 hr after infection by CC17 GBS (n = 9 mice per group). (e to g) Mice were infected intravenously with CC17 GBS ($2.10^7$ CFU, n = 10 mice per group). Bacteremia (e) and total CFU counts in the spleen (f) and brain (g), 2 hr, 24 hr and 48 hr after infection. Red lines are represented at median value. Multiple-group comparisons were performed by non-parametric two-way ANOVA (b) and Kruskal-Wallis test (c to g). *p<0.05; **p<0.01; ***p<0.001; ns: not significant.

DOI: https://doi.org/10.7554/eLife.48772.002

The following figure supplements are available for figure 1:

**Figure supplement 1.** Growth curves of CC17 and CC23 GBS in presence of hormones.

DOI: https://doi.org/10.7554/eLife.48772.003

**Figure supplement 2.** Bacterial counts of CC17 GBS 24 hr following mice oral infection (n = 8 mice per group).

DOI: https://doi.org/10.7554/eLife.48772.004

## CC17 GBS crossing of the intestinal barrier and dissemination is enhanced by E2-P4 $C_7$ hormonal concentrations

In the model of meningitis following oral gavage, several factors may participate to the severity of infection, including the capacity to cross the intestinal barrier, to disseminate, and eventually to cross

the BBB. To identify the steps at which E2-P4 concentrations contribute to CC17 GBS virulence, bacteria were enumerated in the mesenteric lymph nodes (MLN), spleen, and blood, 2 hr and 24 hr following mice oral inoculation. CC17 GBS crossed the intestinal barrier and reached the MLN, the spleen, and the blood within 2 hr after mice gavage (*Figure 1c*). However, no bacteria could be detected 24 hr after infection (*Figure 1—figure supplement 2*), indicating early digestive translocation and dissemination, followed by efficient bacterial clearance. Besides, CC17 bacterial counts in the MLN and in the blood 2 hr after infection were increased ~10 fold in $C_7$ mice in comparison to $C_0$ and control mice (*Figure 1c*), showing that E2-P4 $C_7$ hormonal condition promoted CC17 GBS dissemination. The enhanced invasiveness of CC17 GBS in E2-P4 $C_7$ mice was associated with a stronger inflammatory response compared to $C_0$ and control mice as indicated by the increased circulating levels of several cytokines including the interleukins IL-1ß and IL-10, and the leukocytes chemoattractant proteins CCL20 and CXCL2 which are secreted by epithelial and immune cells, respectively (*Ranasinghe and Eri, 2018*) (*Figure 1d*).

In parallel, we also investigated the possibility of a better survival of CC17 GBS to the host's immune defenses and of an enhanced translocation of the BBB in E2-P4 $C_7$ mice. To this end, we used a murine model in which mice were submitted to the hormonal exposure described above and infected intravenously with $2.10^7$ CFU of the CC17 GBS isolate. The resulting bacteremia (*Figure 1e*) and total CFU counts in the spleen (*Figure 1f*) and brain (*Figure 1g*) were not statistically different between $C_0$, $C_7$, and control mice, 2 hr, 24 hr and 48 hr after infection, showing that neither bacterial clearance nor bacterial crossing of the BBB were affected by the hormonal conditions tested.

## CC17 GBS crosses the intestinal barrier through Peyer's patches in a process favored by E2-P4 $C_7$ concentrations

Our results indicated that the severity of infection in E2-P4 $C_7$ mice was mainly due to an enhanced crossing of the intestinal barrier which occurred quickly after infection, as exemplified by the CFU counts in the MLN 2 hr following mouse oral gavage (*Figure 1c*). Therefore, we sought to identify the portal of entry of CC17 GBS. To this end, we enumerated whole tissue-associated and intratissular bacteria, reflecting adherent and invasive bacteria, respectively, in both individualized Peyer's patches which constitute a portal of entry for gastrointestinal pathogens (*Da Silva et al., 2017*) and in the rest of the small intestine. Following oral infection by CC17 GBS, similar levels of tissue-associated bacteria were observed in $C_0$, $C_7$, and control mice (*Figure 2a*) but a stronger invasion of the intestine and the Peyer's patches was seen in $C_7$ mice (*Figure 2b*), demonstrating an increased crossing of the intestinal barrier by CC17 GBS through the Peyer's patches in E2-P4 $C_7$ hormonal condition.

Peyer's patches are part of the gut associated lymphoid tissue, and the follicle-associated epithelium (FAE) overlying them is specialized for sampling luminal content. This process is achieved by microfold cells, also referred to as M cells, which represent approx. 10% of the cells within the FAE. M cells provide an efficient delivery of microorganisms and antigens to the underlying mucosal tissue, which contains an assorted population of lymphocytes, macrophages and dendritic cells (DC) (*Mabbott et al., 2013*; *Nakamura et al., 2018*). DC can also directly uptake the bacteria from the intestinal lumen by extending their dendrites through the tight junctions between enterocytes or through cellular pores in the M cells (*Mabbott et al., 2013*). Bacterial association with M cells and DC being transient, we chose to perform intestinal ligated loops infections in order to maintain a high GBS density in the intestinal lumen and to increase the probability of observing GBS interactions with host cells. Using confocal microscopy imaging of mouse intestinal ligated loops infected by CC17 GBS, we observed both M cells-associated bacteria (*Figure 2c*) and bacteria associated with CD11c positive cells, CD11c being a marker for macrophages and DC (*Figure 2d–e*). Besides, consistent with the elevated circulating levels of the chemokine CCL20 observed in E2-P4 $C_7$ mice (*Figure 1d*), intestinal mRNA levels of the genes encoding the chemokine CCL20 and its cognate receptor C-C motif chemokine receptor 6 (CCR6) which are involved in leucocytes recruitment to the epithelium were also increased in E2-P4 $C_7$ mice compared to $C_0$ and control mice (*Figure 2f*). In parallel, genes expressed by immune cells and involved in DC migration to secondary lymph nodes, including the chemokines encoding genes *Ccl19* and *Ccl21* and their receptor *Ccr7* involved in DC migration (*Riol-Blanco et al., 2005*), were overexpressed ~2 fold in the intestinal tissue of E2-P4 $C_7$ compared to $C_0$ and control mice (*Figure 2g*). This overexpression of DC activation and migration markers was associated with a higher proportion in the MLN of E2-P4 $C_7$ mice of cells positive for

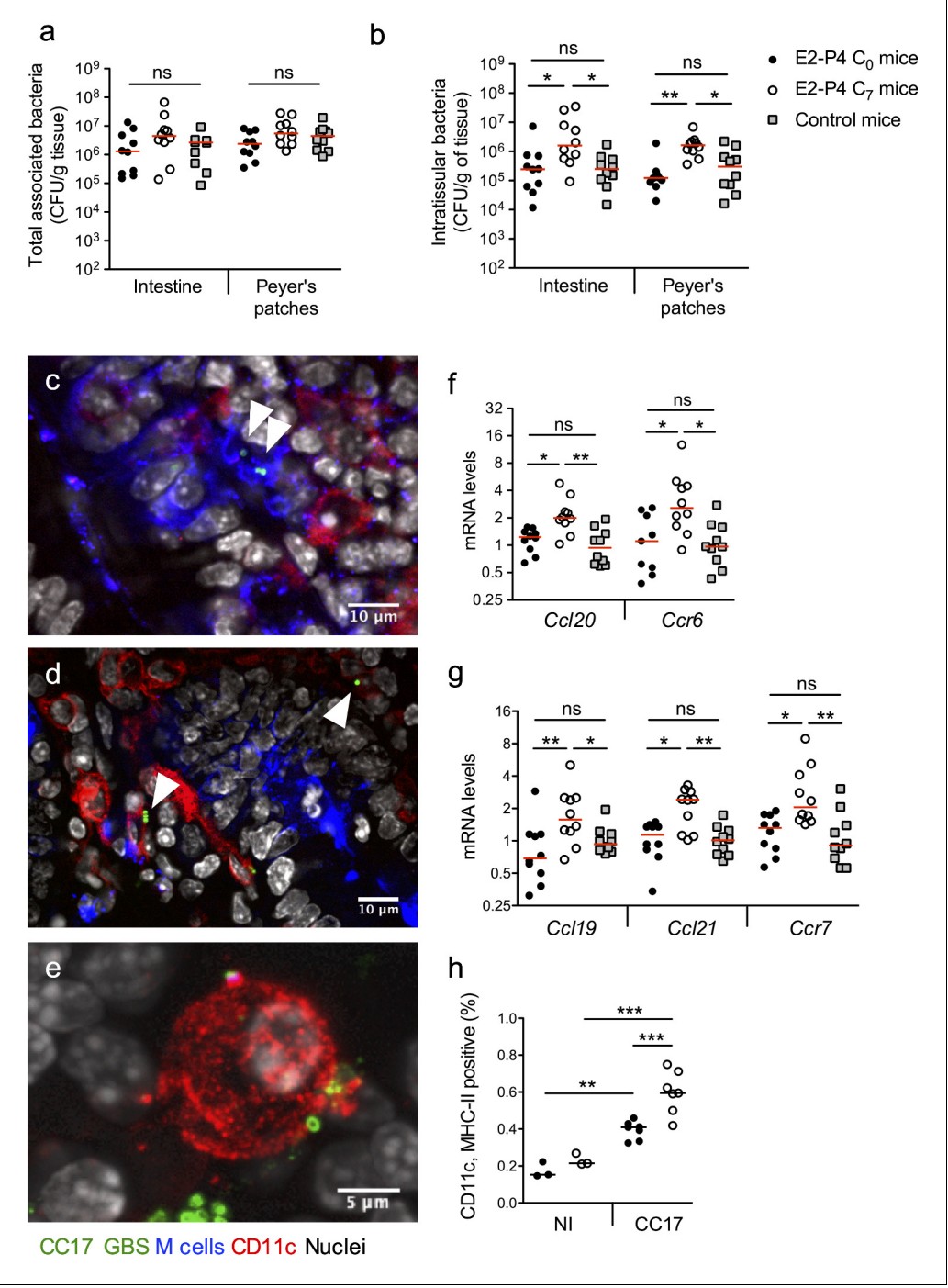

**Figure 2.** E2-P4 $C_7$ concentrations favor CC17 GBS crossing of the intestinal barrier through Peyer's patches. SPF 3-week-old mice were administered E2-P4 cocktails for hormonal impregnation as described in *Figure 1*. (**a, b, and f to h**) Mice were gavaged by $2.10^{10}$ CFU of CC17 GBS. (**a, b**) Bacteria were enumerated 2 hr after infection in the intestine and Peyer's patches either as tissue-associated bacteria (**a**) or as intratissular bacteria following tissue treatment with gentamicin (**b**). Bacterial counts are expressed in CFU per gram of tissue and were obtained following tissue homogenization and plating on Granada medium (n = 10 mice per group). (**c to e**) Representative confocal images (out of 12) of E2-P4 $C_7$ mice small intestine ligated loops infected by $2.10^{10}$ CFU of CC17 GBS for 1 hr and showing intratissular bacteria (n = 3 mice). Small intestine sections showing bacteria associated with M cells (**c**) and CD11c positive cells (**d**). (**e**) View of a tridimensional representation of CC17 GBS associated with a CD11c positive cell. M cells were labeled with anti-UEA-I antibody, dendritic cells (DC) and macrophages with anti-CD11c antibody and nuclei with DAPI. White triangles indicate GBS cocci. (**f, g**) Intestinal mRNA levels of genes

*Figure 2 continued on next page*

*Figure 2 continued*
involved in leukocytes recruitment (f) and DC activation and migration (g) 2 hr after mouse oral gavage. Results are normalized to *Actin* and expressed as mean fold change relative to control condition ± SEM (n = 10 mice per group). (h) Flow cytometry analysis of CD11c and MHC-II positive cells in MLN of non-infected (NI, n = 3 mice per group) and CC17-infected mice (n = 7 mice per group) 2 hr after oral gavage of E2-P4 $C_0$ and E2-P4 $C_7$ mice. (a, b, and f to h) Lines are represented at median value. Multiple-group comparisons were performed by non-parametric two-way ANOVA (a, b) and Kruskal-Wallis test (f to h). *p<0.05; **p<0.01; ***p<0.001; ns: not significant.
DOI: https://doi.org/10.7554/eLife.48772.005

CD11c and class II major histocompatibility complex (MHC-II) which are markers of activated DC (*Figure 2h*). Altogether, these data reflect an enhanced DC maturation and migration in E2-P4 $C_7$ condition correlated to the improved CC17 GBS crossing of the intestinal barrier following oral infection.

## E2-P4 $C_7$ concentrations enhance CC17 GBS transcytosis through M cells

These results suggested that CC17 GBS translocation across the intestinal epithelium occurred in Peyer's patches, likely by transcytosis through M cells. Thus, we addressed the possibility of a direct bacterial crossing of the intestinal barrier and whether E2-P4 $C_7$ hormonal concentrations could favor this invasion route. To this end, we first measured bacterial adherence, invasion, and transcytosis through a monolayer of the Caco-2 enterocyte cell line cultured on filters until complete differentiation and polarization. Adherence and invasion of the CC17 GBS was not modified by E2-P4 concentrations (*Figure 3a*) and the bacterial counts in the lower compartment of the cell culture chamber up to 8 hr after enterocytes infection by their apical side were always below the detection limit, indicating that the bacteria were unable to cross the enterocytes monolayer whatever the hormonal condition tested.

Next, we investigated bacterial ability to translocate through a monolayer composed of both enterocytes and M cells, which can be obtained by co-cultivation of Caco-2 cells with the Raji B lymphocytes cell line (*Gullberg et al., 2000*; *Kernéis et al., 1997*). The differentiation of enterocytes into M cells which are characterized by the lack of a typical enterocyte brush border is attested by the loss of apical actin labeling and by the expression for a subset of them of the M cell-specific surface glycoprotein 2 (GP2) (*Figure 3b*), which reflects M cell full differentiation and better transcytosis capacities (*Kanaya et al., 2018*; *Kimura et al., 2015*). Using GP2 labeling, the percentage of mature M cells was estimated at approx. 5% for each experiment.

In this co-culture system, CC17 GBS could cross the epithelial layer, as assessed by the percentage of transcytosis obtained 2 hr after infection (*Figure 3c*). Furthermore, the transcytosis was potentiated approx. 2-fold in E2-P4 $C_7$ condition in comparison to $C_0$ and control (no hormones) conditions, without being associated with increased cytotoxicity or alterations of the epithelial barrier junctions, as measured by lactate dehydrogenase release and by the transport of the Lucifer yellow fluorescent dye (*Putt et al., 2017*), respectively (data not shown). We then used immunofluorescence imaging and quantified the association between the bacteria and the cells constituting the epithelial layer, that is enterocytes on one hand and GP2 positive M cells on the other hand (*Figure 3d–h* and *Figure 3—figure supplement 1*). Thereby, we confirmed as previously described (*Tazi et al., 2010*) that CC17 GBS was more adherent to intestinal epithelial cells than CC23 GBS (*Figure 3d–e*), and more interestingly, we observed that approx. 50% of the cell-associated CC17 GBS were actually associated with M cells, either as adherent or intracellular bacteria (*Figure 3f–g*), in contrast to CC23 GBS, which were at 90% associated with enterocytes (*Figure 3f* and *Figure 3—figure supplement 1*). Besides, the proportion of M cells-associated CC17 GBS was ~50% higher in E2-P4 $C_7$ than in $C_0$ and control conditions (*Figure 3h*) despite similar proportion of M cells and similar adherence to the enterocytes + M cells epithelial monolayer in the three conditions (data not shown).

## E2-P4 concentrations modulate M cells differentiation

The enhanced transcytosis observed in E2-P4 $C_7$ condition associated with the increased interaction between CC17 GBS and M cells could be linked to variations in M cells differentiation. Intestinal

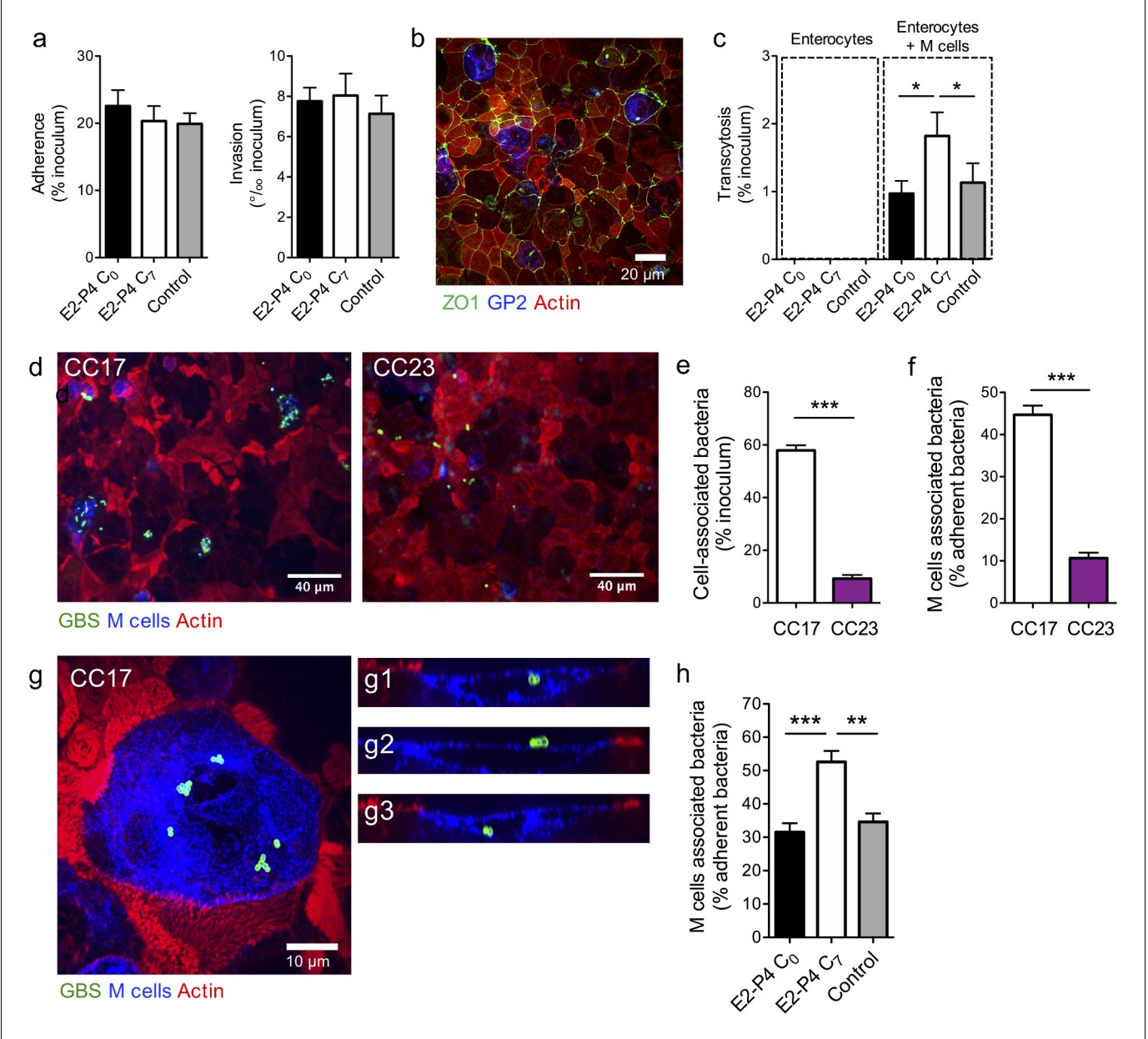

**Figure 3.** E2-P4 $C_7$ concentrations favor CC17 GBS transcytosis through M cells. Human Caco-2 enterocytes were polarized on standard plates (**a**) or Transwell inserts (**b to h**) for 14–21 days. (**b to h**) Human Raji B lymphocytes were added in the lower compartment of the culture chamber in order to obtain a cellular layer composed of enterocytes and M cells and the cell culture medium was supplemented with E2-P4 concentrations equivalent to those found at birth (E2: $10^{-8}$M, P4: $10^{-6}$M; E2-P4 $C_0$ condition) and 7 days later (E2: $10^{-9}$M, P4: $10^{-7}$M; E2-P4 $C_7$ condition) for 6 days before infection. Control cells were maintained in regular cell culture medium without E2 and P4 supplementation. Epithelial cells were infected by their apical side with bacteria at a multiplicity of infection 10 (**a**) and 100 (**c to h**) for 2 hr. (**a**) Bacterial adherence and invasion of a cellular monolayer composed of enterocytes. (**b**) Representative confocal microscopy image of the epithelial layer obtained following Caco-2 and Raji co-culture showing enterocytes and M cells. (**c**) Transcytosis of CC17 GBS across a monolayer strictly composed of enterocytes and a monolayer composed of both enterocytes and M cells. (**d**) Representative confocal microscopy images showing CC17 and CC23 GBS adherence to the enterocytes + M cells epithelial layer. (**e**) Bacterial adherence of CC17 and CC23 GBS to the enterocytes + M cells epithelial layer. (**f**) Quantitative imaging analysis of bacterial association with M cells expressed as the proportion of bacteria associated with GP2 positive cells relative to total adherent bacteria. (**g**) Representative confocal microscopy image showing the presence of several CC17 GBS cocci at the surface and inside an M cell. (**g1 to g3**) Orthogonal views of (**g**). (**d to g**) Experiments were carried out in E2-P4 $C_7$ condition. (**h**) Quantitative imaging analysis of CC17 GBS association with M cells in E2-P4 $C_0$, E2-P4 $C_7$, and control condition. (**a, c, e, f, h**) Results are expressed as mean ± SEM.≥3 experiments in duplicate. (**b, d, g**) Tight junctions are labeled with anti-ZO1 antibody, M cells with anti-GP2 antibody, GBS with anti-GBS antibody, and actin with phalloidin. Multiple-group comparisons were performed by Kruskal-Wallis test (**c and h**) and single-group comparisons by Mann-Whitney test (**e and f**). *p<0.05; **p<0.01, ***p<0.001.

*Figure 3 continued on next page*

*Figure 3 continued*

DOI: https://doi.org/10.7554/eLife.48772.006

The following figure supplement is available for figure 3:

**Figure supplement 1.** Confocal microscopy imaging of GBS interactions with enterocytes and M cells *in vitro*.

DOI: https://doi.org/10.7554/eLife.48772.007

stem cells differentiation into M cells is under the control of exogenous factors including receptor activator of NF-KB ligand (RANKL) which is expressed by stromal cells and lymphocytes underneath the FAE (*Knoop et al., 2009*; *Nagashima et al., 2017*). Ligation of RANKL to its cognate receptor RANK at the surface of epithelial cells activates the NF-KB pathway and induces the expression of the transcriptional factor Spi-B which in turn activates the expression of M cell-specific molecules such as GP2 (*Kanaya et al., 2018*).

Therefore, we measured the expression of *RANKL* in the Raji B lymphocytes used for M cells differentiation *in vitro* and found that the mRNA levels were three times higher in E2-P4 $C_7$ condition than in E2-P4 $C_0$ and control conditions (*Figure 4a*). In parallel, in epithelial cells, mRNA levels of Spi-B and of its target genes involved in M cells differentiation (*TNFAIP2*, *GP2*) were also increased 2- to 3-fold in E2-P4 $C_7$ condition (*Figure 4b*) and GP2 expression at least doubled both at the transcriptional and at the protein level (*Figure 4b–d*). The expression of genes involved in M cells differentiation was then measured in the Peyer's patches of mice submitted to E2-P4 $C_0$ or $C_7$ hormonal

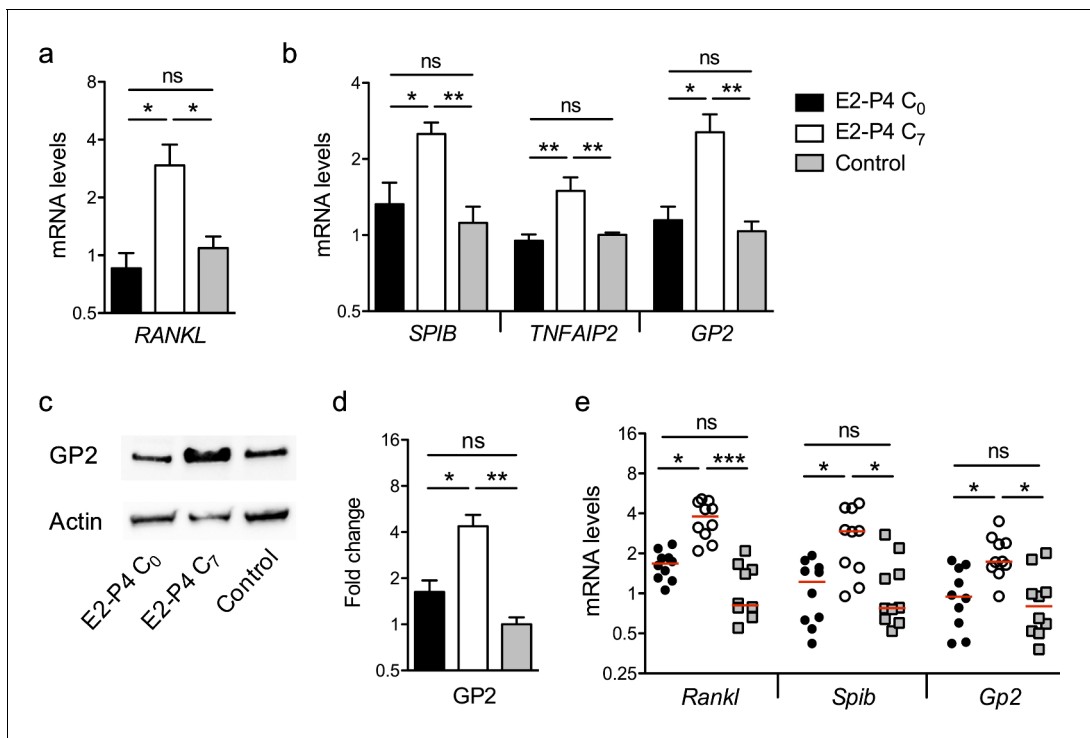

**Figure 4.** E2-P4 $C_7$ concentrations promote intestinal M cells maturation. (a–d) The epithelial layer composed of enterocytes and M cells was prepared as described in *Figure 3*. Following 6 days of Caco-2 and Raji co-culture, lymphocytes and epithelial cells were recovered and lysed for gene and protein expression analysis. (a) Raji B lymphocytes mRNA levels of *RANKL*. (b) Caco-2 mRNA levels of Spi-B-dependent M cells differentiation genes. (a–b) Results are normalized to *ACTIN* and expressed as mean fold change relative to control condition ± SEM. (c–d) Western blot analysis of GP2. (c) Representative sample blot and (d) quantification of GP2 protein levels. Values are presented as the ratio between GP2 and Actin and expressed as mean fold change relative to control condition ± SEM. ≥3 experiments in duplicate. (e) SPF 3-week-old mice were administered E2-P4 cocktails as described in *Figure 1*. At the end of the hormonal treatment, Peyer's patches were recovered for mRNA quantification of M cells differentiation markers. Results are normalized to *Actin* and expressed as mean fold change relative to control condition ± SEM (n = 10 mice per group). *p<0.05, **p<0.01, ***p<0.001, ns: not significant (Kruskal-Wallis test).

DOI: https://doi.org/10.7554/eLife.48772.008

impregnation and in control mice. In agreement with the results obtained *in vitro*, mRNA levels of murine *Rankl*, *Spib* and *Gp2* were increased 2- to 3-fold in $C_7$ condition compared to $C_0$ and control condition (*Figure 4e*). Altogether, our results indicate that M cells differentiation is influenced by E2-P4 hormonal concentrations and increased in E2-P4 $C_7$ condition.

### The surface protein Srr2 contributes to CC17 GBS interaction with M cells and to the severity of meningitis in E2-P4 $C_7$ condition

The specific association between CC17 GBS and M cells (*Figure 3f*) indicated that a virulence factor specifically expressed by CC17 GBS was involved in the bacterial – cell interaction and prompted us to investigate the role of the two CC17 GBS surface proteins HvgA and Srr2 in this interaction. Indeed, molecular epidemiological data using either gene sequencing or whole genome sequencing revealed that HvgA and Srr2 are always and almost exclusively present in CC17 strains, whereas non-CC17 isolates express other allelic variants of these proteins. Both HvgA and Srr2 have been shown to play a critical role in CC17 GBS virulence (*Brochet et al., 2006*; *Da Cunha et al., 2014*; *Six et al., 2015*; *Tazi et al., 2010*). Using $\Delta srr2$ and $\Delta hvgA$ inactivated mutants, we first analyzed bacterial adherence to the enterocytes + M cells monolayer and showed that both HvgA and Srr2 contribute to GBS adherence to the intestinal epithelial cells in E2-P4 $C_7$ condition (*Figure 5a*). Nevertheless, only Srr2 was involved in CC17 GBS association with M cells as objectified by quantitative image analysis which showed a 50% reduction in the association of the $\Delta srr2$ mutant with GP2 positive cells in comparison to the wild-type (WT) strain and to the reverted mutant (back to WT, BTWT) isolate (*Figure 5b*). In agreement with these results, the transcytosis capacity of the $\Delta srr2$ mutant across the enterocytes + M cells monolayer was drastically impaired (*Figure 5c*).

These *in vitro* observations pointed at Srr2 as a major determinant of CC17 GBS intestinal barrier crossing, particularly in hormonal conditions mimicking those of >7 day-old neonates. To confirm this hypothesis, we addressed bacterial translocation capacities in E2-P4 $C_0$ and E2-P4 $C_7$ mice and quantified the bacteria in the Peyer's patches, MLN, and brain 2 hr after oral infection by the WT CC17 and its $\Delta srr2$ mutant (*Figure 5d–g*). The results clearly indicate that Srr2 contributes to CC17 GBS association with Peyer's patches (*Figure 5d*), to Peyer's patches invasion (*Figure 5e*), to dissemination to the MLN (*Figure 5f*), and to central nervous system invasion (*Figure 5g*). Furthermore, in contrast to infection by the WT CC17 GBS, infection by the $\Delta srr2$ mutant did not lead to higher bacterial counts in $C_7$ mice than in $C_0$ mice (*Figure 5c–f*), suggesting that Srr2 participates to the increased virulence of CC17 GBS in E2-P4 $C_7$ hormonal condition.

## Discussion

Steroid hormones display pleiotropic effects on host's physiology and strongly modulate physiological barriers permeability and the immune response to pathogens (*Robinson and Klein, 2012*). However, despite neonates' massive intrauterine exposure to E2 and P4 and their obvious contribution to immune system regulation and to the outcome of many infectious diseases during pregnancy, their impact on neonates' susceptibility to infection remains poorly studied and is often restricted to investigating alterations due to hormone levels found in cord blood (*Giannoni et al., 2011*; *Kraft et al., 2013*). Yet, E2 and P4 circulating concentrations remain elevated up to 6 months of age (*Khashana et al., 2016*; *Kuijper et al., 2013*), and using models representative of concentrations encountered in a newborn and in a neonate aged over 7 days old, we could i) unravel the role of M cells which are found in the Peyer's patches as major portals of entry of CC17 GBS invasive infections and ii) demonstrate that hormonal levels regulate M cells maturation and therefore CC17 GBS invasiveness. Indeed, M cells are necessary for CC17 GBS crossing of intestinal epithelial cells *in vitro*, indicating that the preferential invasion route for CC17 GBS in this cellular model is transcytosis through M cells. Transcytosis is potentiated by E2-P4 hormonal conditions encountered in over 7 day-old neonates in correlation with a better M cell maturation *in vitro* and *in vivo* and a higher CC17 GBS association with M cells *in vitro*. Accordingly, CC17 GBS crossing of the intestinal barrier *in vivo* including CC17 GBS invasion of the Peyer's patches is increased by E2-P4 hormonal conditions encountered in over 7 day-old neonates. Thus, we identified a cellular pathway for CC17 GBS intestinal translocation and showed that CC17 GBS exploits M cells to invade the host and reach the lymphatic system to disseminate. Importantly, we also demonstrate that GBS – M cells interaction is specific to the hypervirulent CC17 clone and mediated by the surface protein Srr2 revealing a new

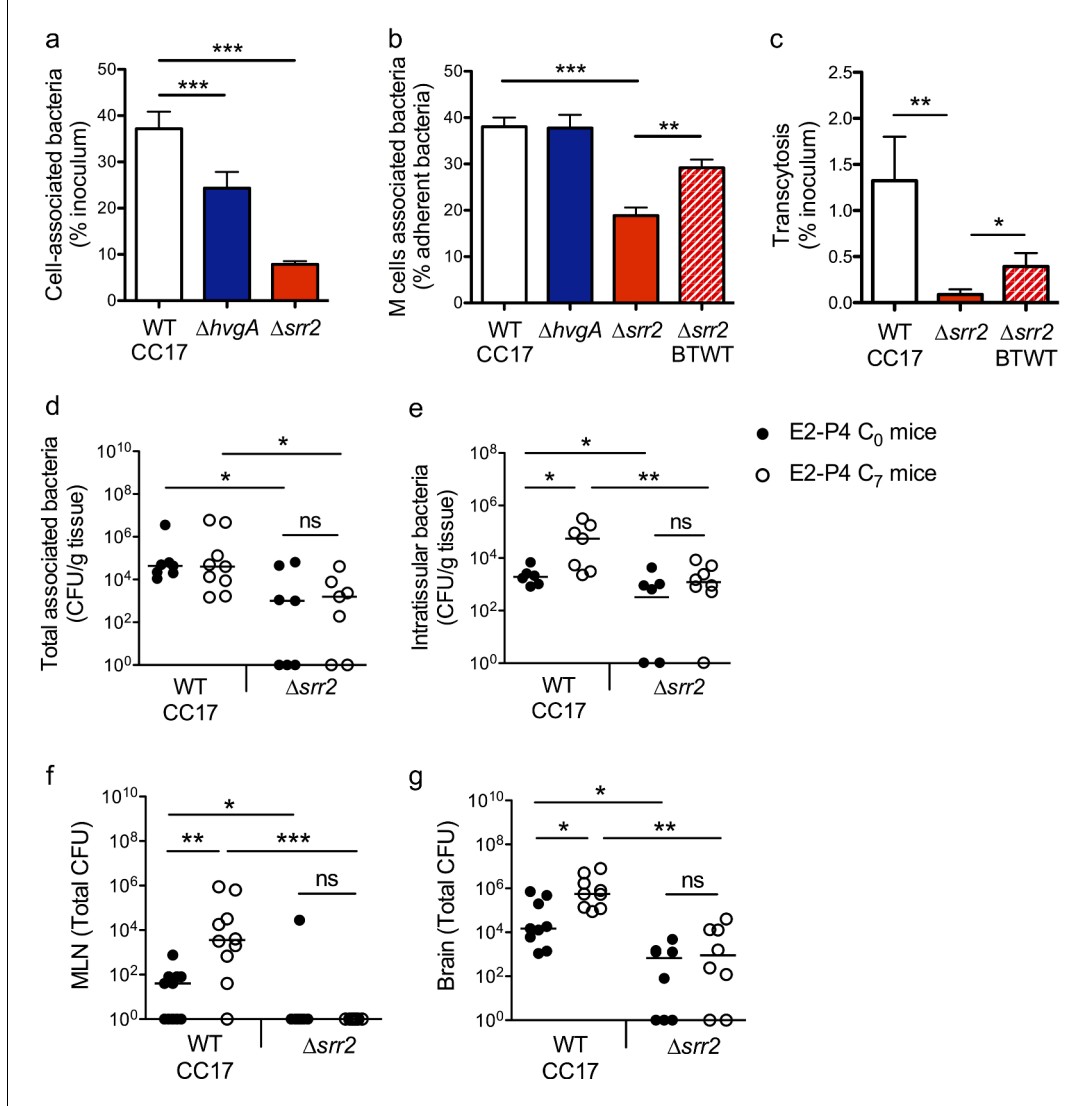

**Figure 5.** The CC17 surface protein Srr2 promotes GBS interaction with M cells and contributes to CC17 GBS hypervirulence in E2-P4 $C_7$ condition. (a to c) The epithelial monolayer composed of enterocytes and M cells was prepared as described in *Figure 3* and cultured in E2-P4 $C_7$ condition. Lymphocytes were removed before infection of the upper compartment with bacteria (multiplicity of infection 100). Cell-associated bacteria were enumerated 2 hr after infection and cells were fixed for staining. (a) Adherence of the wild-type (WT) CC17 GBS and its Δ*hvgA* and Δ*srr2* mutants. (b) Quantitative imaging analysis of bacterial association with M cells of the WT CC17 GBS, the Δ*hvgA* and Δ*srr2* mutants, and the Δ*srr2* back to WT (BTWT) reverted mutant. (c) Transcytosis of the WT CC17 GBS, the Δ*srr2* mutant, and the BTWT across the enterocytes + M cells monolayer. Results are expressed as mean ± SEM.≥3 experiments in triplicate. (d to g) SPF 3-week-old mice were administered E2-P4 cocktails as described in *Figure 1* before being gavaged with the WT CC17 GBS or its Δ*srr2* mutant ($2.10^{10}$ CFU). Bacteria were enumerated 2 hr after infection in the Peyer's patches either as tissue-associated bacteria (d) or as intratissular bacteria following tissue treatment with gentamicin (e), in the MLN (f), and in the brain (g). (d to f) n = 10 mice per group. (g) n = 7–9 mice per group. (d and e) Results are expressed as total CFU per gram of tissue. (f and g) Results are expressed as total CFU per organ. Lines are represented at median values. *p<0.05; **p<0.01; ***p<0.001; ns: not significant (Kruskal-Wallis test).
DOI: https://doi.org/10.7554/eLife.48772.009

contribution of this virulence factor to CC17 GBS pathogenesis. Because the surface protein Srr2 is always and almost exclusively present in GBS isolates belonging to the highly homogenous hypervirulent CC17 clone (*Brochet et al., 2006*; *Da Cunha et al., 2014*), and because CC17 GBS invasiveness depends on its ability to cross the intestinal barrier through M cells, our observations that E2 and P4 concentrations encountered in over 7 day-old neonates favor CC17 GBS infection must apply to all CC17 GBS strains. Besides, although being impaired but not totally abolished, the capacity of the Δ*srr2* mutant to cross the intestinal barrier and to reach the central nervous system is not

enhanced in E2-P4 hormonal conditions encountered in over 7 day-old neonates, suggesting that the increased virulence of the WT CC17 in this condition is Srr2 dependent.

We previously showed that CC17 GBS was able to cause invasive disease and meningitis following oral infection, thereby demonstrating its ability to cross the intestinal barrier (*Tazi et al., 2010*). This infection route was furthermore associated with a dose-dependent mortality. Notably, mouse oral gavage with $1.10^{10}$ CFU was associated with 60% mortality in the 12 hr following infection. Conversely, in this work, mice gavage with $2.10^{10}$ CFU did not lead to any mortality for up to 24 hr in any of the 3 groups of mice, indicating that the difference between these two models was not attributable to the hormonal treatments but more likely to the age of the mice at the time of infection. Indeed, in our previous work, CC17 GBS ability to cross the intestine was assessed in 2 to 3 week-old pre-weaning mice whereas in this work, experiments were performed with 3 week-old mice which were stimulated with a hormonal treatment for almost one week and which were eventually 4 weeks old at the time of infection, emphasizing the age-related susceptibility to infection. Another interesting feature of the animal models used in this work resides in CC17 GBS capacity to reach the brain more efficiently in the first hours following oral than intravenous infection despite similar levels of bacteremia in both models (*Figure 1*). This difference gives insights into the dissemination pathways taken by GBS following each of these infection modes. After intravenous injection in the tail vein, bacteria reach the inferior vena cava (*Cook, 1965*) and the vast majority of them is then captured by the spleen and the liver. Conversely, when bacteria disseminate through the lymphatic vessels after they got to the MLN following oral infection, they eventually reach the blood circulation *via* the thoracic duct directly in the subclavian vein, thereby shunting the hepatic portal system (*Hsu and Itkin, 2016*). These major differences are objectivized by the bacterial loads in the spleen following these 2 modes of infection (*Figure 1*) and reinforce the hypothesis that CC17 GBS preferentially disseminates *via* the lymphatic system following oral infection.

The overwhelming responsibility of CC17 GBS in neonatal invasive disease especially in LOD and meningitis has been attributed to the specific expression of the surface proteins HvgA and Srr2 (*Six et al., 2015*; *Tazi et al., 2010*). Indeed, comparative genomics revealed that CC17 strains, which constitute a highly homogenous clonal complex (*Da Cunha et al., 2014*), exhibit a few specific genes (*Brochet et al., 2006*). Among these, *hvgA* and *srr2* represent the only surface proteins encoding genes that are always and almost exclusively expressed by the hypervirulent clone. Both proteins promote CC17 GBS adherence to a wide variety of cells including brain endothelial cells and are involved in BBB crossing (*Seo et al., 2013*; *Six et al., 2015*; *Tazi et al., 2010*), leading to the conclusion that HvgA and Srr2 account for the meningeal tropism of the hypervirulent clone. Besides, HvgA confers gut colonization capacities in murine models and it was presumed that CC17 GBS association with LOD was related to an increased ability of CC17 GBS to colonize neonates compared to other GBS clones (*Tazi et al., 2010*). However, a recent large prospective cohort study of infant colonization by GBS showed that instead of colonizing neonates more efficiently than other GBS clones, CC17 GBS was associated with a higher risk of delayed post-delivery colonization (*Tazi et al., 2019*). Together with our findings, these results suggest that the important responsibility of CC17 GBS in LOD relies i) on its improved capacity to establish in the infant developing and versatile microbiota, advantage which is likely conferred by HvgA, and ii) on Srr2-mediated ability to use M cells to cross the intestinal barrier and eventually to disseminate.

Enteroinvasive pathogens have developed numerous strategies to subvert M cells and gain access to deeper tissues (*Da Silva et al., 2017*; *Nakamura et al., 2018*). These strategies include direct binding of bacterial effectors with M cell-specific carbohydrates and receptors, such as the FimH pilus protein of *Escherichia coli* and *Salmonella enterica* serovar Typhimurium interaction with the GP2 M cell surface protein (*Mabbott et al., 2013*), but also the direct regulation of M cell maturation (*Tahoun et al., 2012*). Hence, CC17 GBS interaction with M cells might involve an M cell-specific receptor targeted by Srr2. Besides, E2 and P4 contribute to CC17 transcytosis by regulating M cell maturation, likely through alterations of the NF-KB pathway which is known to be regulated by steroid hormones and directly involved in M cells differentiation (*Kanaya et al., 2018*; *Robinson and Klein, 2012*). Whether the differentiation process strictly depends on E2 or P4 or on a combination of both hormones still has to be investigated. At last, steroid hormones impact the immune responses and might thereby influence microbiota composition which in turn shapes the immune system development and the maturation of the intestinal epithelium (*Sommer and Bäckhed, 2013*). Therefore, gut microbiota and GBS colonization likely contribute to M cells maturation and further

studies are needed to address the possible relationship between microbiota composition, intestinal colonization by the different GBS clones, and the onset of LOD.

Finally, the rapidity of CC17 GBS entire infectious process which takes place within 2 hr from mice oral infection to central nervous system invasion together with the low-associated bacteremia gives rise to the hypothesis of a dissemination through a Trojan horse mechanism which is exacerbated in E2-P4 $C_7$ condition. This hypothesis is supported by the association of CC17 GBS with CD11c - a marker for macrophages and DC - positive cells in the Peyer's patches and by the increased proportion of CD11c positive cells in the MLN of CC17-infected mice, especially in E2-P4 $C_7$ condition. Indeed, enteric pathogens can invade the host using various cellular routes including transcytosis across M cells, transcytosis across enterocytes or direct uptake from the intestinal lumen by Peyer's patches DC which can extend their dendrites through M cells or between enterocytes tight junctions (*Lelouard et al., 2012*; *Mabbott et al., 2013*). Once in the *lamina propria*, bacteria can be taken up by mononuclear phagocytes, including macrophages and other DC subsets. Unlike macrophages that rapidly kill microorganisms in their phagosomes, classical DC can retain unprocessed antigens, including bacteria, for several days before transferring these antigens to naive B cells to initiate a specific antibody response. Hence, viable commensal bacteria can be recovered from classical DC migrating within afferent mesenteric lymph and this process aiming at initiating the mucosal immune response can be exploited by some pathogens to disseminate. For instance, *Toxoplasma gondii* infects intestinal DC and subverts their migration capacities to gain access to the brain (*Courret et al., 2006*). Whether CC17 GBS can be taken up by intestinal DC, survive in these cells hidden from the host immune system, and subvert their migration capacities to reach the central nervous system still has to be investigated.

In conclusion, our results provide new insights for the understanding of CC17 GBS pathophysiology and for its intimate association with neonatal LOD, which can now not only be linked to HvgA-mediated intestinal colonization (*Tazi et al., 2010*), but also to an efficient Srr2-mediated interaction with M cells conditioned by neonatal hormonal concentrations and gastrointestinal tract development. Given the critical role played by Srr2 in the initiation of the infectious process and the burden of CC17 GBS LOD, Srr2 stands together with HvgA as a highly relevant target for vaccine development (*Lin et al., 2018*; *Santi et al., 2009*; *Six et al., 2015*).

## Materials and methods

**Key resources table**

| Reagent type (species) or resource | Designation | Source or reference | Identifiers | Additional information |
|---|---|---|---|---|
| Strain, strain background (*Streptococcus agalactiae*) | BM110 | *Musser et al., 1989* | N/A | Isolated from patient |
| Strain, strain background (*Streptococcus agalactiae*) | BM110Δ*hvgA* | *Tazi et al., 2010* | CCH395 | |
| Strain, strain background (*Streptococcus agalactiae*) | BM110Δ*srr2* | This study | CCH2040 | Constructed by in-frame deletion (see Materials and methods, section Bacterial strains) |
| Strain, strain background (*Streptococcus agalactiae*) | BM110 BTWTΔ*srr2* | This study | CCH2041 | (see Materials and methods, section Bacterial strains) |
| Strain, strain background (*Streptococcus agalactiae*) | NEM316 | *Glaser et al., 2002* | N/A | Isolated from patient |

*Continued on next page*

*Continued*

| Reagent type (species) or resource | Designation | Source or reference | Identifiers | Additional information |
|---|---|---|---|---|
| Cell line (*Homo sapiens*) | Caco-2/TC7 Colonic epithelial cell | *Chantret et al., 1994* | RRID:CVCL_0233 | |
| Cell line (*Homo sapiens*) | Raji B lymphocytes | ATCC | CCL-86; RRID:CVCL_0511 | |
| Antibody | anti-group B *Streptococcus* (Rabbit polyclonal) | ThermoFisher | PA1-7250; RRID:AB_561572 | IF (1:100) |
| Antibody | anti-group B *Streptococcus* (Mouse serum) | This study | N/A | Homemade (see Materials and methods, section Reagents) IF (1:100) |
| Antibody | anti-human glycoprotein 2 (Rabbit polyclonal) | Abcam | ab129414; RRID:AB_11155051 | IF (1:100) |
| Antibody | anti-human glycoprotein 2 (Rabbit polyclonal) | Abcam | ab105503, RRID:AB_10859272 | WB (1:300) |
| Antibody | anti-mouse CD11c-Alexa Fluor 488 (Armenian hamster monoclonal) | Biolegend | 117311; RRID:AB_389306 | IF (1:100) |
| Antibody | anti-mouse CD11c-Alexa Fluor 647 (Armenian hamster monoclonal) | Biolegend | 117312; RRID:AB_389328 | IF (1:100) |
| Antibody | anti-mouse CD11c-Brilliant Violet 785 (Armenian hamster monoclonal) | Biolegend | 117335, RRID:AB_11219204 | FACS (5 µL per test) |
| Antibody | anti-mouse F4/80-PE/Cy5 (Rat monoclonal) | Biolegend | 123111, RRID:AB_893494 | FACS (5 µL per test) |
| Antibody | anti-mouse CD19-Brilliant Violet 711 (Rat monoclonal) | Biolegend | 115555, RRID:AB_2565970 | FACS (5 µL per test) |
| Antibody | anti-mouse CD45-APC (Rat monoclonal) | Biolegend | 103111, RRID:AB_312976 | FACS (5 µL per test) |
| Antibody | anti-mouse I-A/I-E (anti-MHC-II)-FITC (Rat monoclonal) | Biolegend | 107605, RRID:AB_313320 | FACS (5 µL per test) |
| Chemical compound, drug | UEA-I-Rhodamine | Vector Laboratories | RL-1062, RRID:AB_2336769 | IF (1:200) |
| Chemical compound, drug | Estradiol | Sigma-Aldrich | E2758; CAS 50-28-2 | |
| Chemical compound, drug | Progesterone | Sigma-Aldrich | P8783; CAS 57-83-0 | |
| Software, algorithm | Kaluza | Beckman Coulter | RRID:SCR_016182 | |
| Software, algorithm | Image J | http://imagej/net | RRID:SCR_003070 | |

## Animals

Work on animals was performed in compliance with French and European regulations on care and protection of laboratory animals (European Council Directive 2010/63, French Law 2013–118, February 6, 2013). Animals were housed in the Institut Cochin animal facility accredited by the French Ministry of Agriculture for performing experiments on live rodents. All experiments were approved by the Ethics Committee. Animals were not involved in any previous procedure.

SPF (3-week-old females BALB/c, Janvier, France) were randomly assigned to three experimental groups and administered a hormonal treatment by subcutaneous injection daily. Hormones were dissolved in sesame oil. For the E2-P4 $C_0$ group of mice, hormonal treatment started with one injection of E2 0.4 mg/mL and P4 10 mg/mL the first day, followed by injection of E2 2 mg/ml and P4 66 mg/mL daily for 4 days. For the E2-P4 $C_7$ group of mice, hormonal treatment consisted of the injection of E2 0.4 mg/mL and P4 10 mg/mL daily for 4 days. Control mice were administered sesame oil alone. E2 and P4 doses administered to obtain the desired serum concentrations in E2-P4 $C_0$ and E2-P4 $C_7$ mice were adapted from previous studies (*Cai et al., 2014*; *Hata et al., 2014*). Injection volume of 100 µL was delivered subcutaneously without prior sedation of the mice.

Four hours after the last hormonal injection, mice were administered GBS ($2.10^{10}$ CFU in 300 µL PBS containing 50 mg/mL of $CaCO_3$ by intra-gastric gavage with a stainless-steel tube without prior sedation. For intravenous infections, mice were administered GBS ($2.10^7$ CFU in 200 µL PBS) by injection in the tail vein. At the end of the experiments, mice were killed by cervical dislocation, the blood, spleen, brain, and MLN and small intestine from jejunum to terminal ileum in oral infections recovered and immediately used for measurement of serum parameters (cytokines, E2 and P4 dosage), RNA extraction (small intestine), and bacterial counts. Experiments were carried out at least twice.

For intestinal ligated loops infection, mice (n = 3) were anesthetized with an intraperitoneal injection of 100 µL of xylazine (100 mg/kg) and ketamine (100 mg/kg) and maintained at 37°C for the duration of the experiment. Sections of the small intestine (1.5 cm long) containing at least one Peyer's patch were closed at both ends with surgical sutures (2 to 3 sections per animal). Bacteria ($2.10^{10}$ CFU in 200 µL PBS) were injected in the lumen. Mice were sacrificed 1 hr after infection, the ligated loops were excised and opened longitudinally for washing, fixation and staining.

## Bacterial strains

GBS strain BM110, capsular serotype III, and multi-locus sequence-type clonal complex CC17, and GBS strain NEM316, capsular serotype III, and multi-locus sequence-type clonal complex CC23 are well-characterized and whole-genome sequenced isolates from neonates with invasive infections (*Glaser et al., 2002*; *Musser et al., 1989*). The BM110 in-frame deletion mutant of *hvgA* was previously characterized (*Tazi et al., 2010*). A BM110 in-frame deletion mutant of *srr2* was constructed by double crossing-over with a deleted copy of the gene as previously described (*Six et al., 2015*; *Tazi et al., 2010*), leading to the selection of the mutant or the BTWT complemented strain. Bacteria were cultured at 37°C in Todd Hewitt broth or agar. Bacterial growth was determined using overnight cultures in Todd Hewitt broth diluted 1/100 in Todd Hewitt broth supplemented with desired E2-P4 concentrations and measurements of the optical density at 600 nm.

Bacteria in mid-exponential growth phase ($OD_{600nm}$=0.5) were harvested by centrifugation (10 min at 4,000 g), washed twice with PBS buffer, and re-suspended either in cellular growth medium without FCS supplemented with E2 and P4 at the desired concentrations for cellular experiments, or in PBS for animal experiments.

## Cell lines

The human colonic epithelial cell line Caco-2/TC7 (*Chantret et al., 1994*) (RRID:CVCL_0233) expressing the structural and functional characteristics of mature enterocytes lining the small intestine was cultured at 37°C in a 10% $CO_2$ atmosphere in DMEM high glucose with GlutaMax (Thermo-Fisher) supplemented with pyruvate (ThermoFisher) and non-essential amino-acids (ThermoFisher) with 20% inactivated FCS (Eurobio). Cells were seeded on standard Costar plates (ThermoFisher) or on Corning Transwell plates with a 3 µm polycarbonate filter (VWR) coated with Matrigel (BD Bioscience) at a density of $2 \times 10^5$/cm². The medium was changed every 2–3 days until complete differentiation (14–21 days).

The human B lymphocytes cell line Raji (ATCC CCL-86, RRID:CVCL_0511) was cultured at 37°C in a 5% $CO_2$ atmosphere in RPMI 1640 medium with GlutaMax (ThermoFisher) with 10% inactivated FCS (Eurobio). For Caco-2 differentiation in M cells, Caco-2 cells were seeded on Transwell plates as described above and after 14 days of differentiation, Raji cells were added to the lower compartment of the culture chamber at a density of $3.10^5$/mL in Caco-2 culture medium added with E2 and P4 concentrations equivalent to cord blood ($C_0$: E2 10 nM and P4 1 µM) or to 7 day-old neonates

($C_7$: E2 1 nM and P4 100 mM). The cell culture medium of the upper compartment was also replaced by fresh medium containing the same hormonal concentrations, and replaced every 2–3 days for a total of 6–7 days of Caco-2 - Raji co-culture. Raji cells were removed and Caco-2 cells were washed twice before infection.

Cell viability was monitored using the lactate dehydrogenase cytotoxicity assay kit (Thermo-Fisher). Cell permeability was monitored using the Lucifer Yellow dipotassium salt (Sigma-Aldrich). Cell lines identity has been authenticated using STR profiling. Cells were tested for mycoplasma using the MycoAlert mycoplasma detection kit (Lonza) every 2 weeks and always found to be negative.

## Bacterial CFU counts

Organs were homogenized in 2 mL (brain, spleen, intestine, Peyer's patches, MLN) or 7 mL tubes (liver) containing a mix of 1.4 and 2.8 mm glass beads and sterile water using the Precellys system (Bertin Technologies). For quantification of intratissular bacteria, small intestine and Peyer's patches were treated with 200 mg/L gentamicin for 2 hr at room temperature to kill extratissular bacteria before being washed three times in PBS and homogenized. Cultured cells were lysed with ice-cold water.

Serial dilutions were plated on Todd Hewitt medium (Becton Dickinson), and on Granada medium (bioMérieux) for intestine, Peyer's patches, and MLN. Plates were incubated 24 hr at 37°C in aerobic (Todd Hewitt medium) or anaerobic (Granada medium) atmosphere.

## Histological procedures and staining

Intestinal ligated loops were excised, flushed, opened longitudinally, and fixed in a periodate–lysine–paraformaldehyde (PFA) solution (0.05 M phosphate buffer containing 0.1 M L-lysine (pH 7.4), 2 mg/mL $NaIO_4$, and 10 mg/mL PFA) overnight at 4°C. Tissues were then embedded in agarose and cut in sections of 200 µm. Permeabilization was performed with saponin 0.5% for 5 min, followed by treatment with bovine serum albumin (BSA) (3% in PBS) for 30 min before immune staining.

Cultured cells were fixed in PFA (4% in PBS) 20 min at room temperature, treated with $NH_4Cl$ (50 mM in PBS) 5 min, permeabilized 5 min in Triton X100 (0.1% in PBS), then treated with BSA (3% in PBS) 30 min.

Actin was labeled with Alexa Fluor 647 phalloidin (1:40, ThermoFisher A22287, RRID:AB_2620155), GBS with rabbit anti-GBS antibody (1:100, ThermoFisher PA1-7250, RRID:AB_561572) or mouse serum anti-GBS (1:100, homemade), human M cells with rabbit anti-GP2 (1:100, abcam ab129414, RRID:AB_11155051) and murine M cells with rhodamine conjugated UEA-I (1:200, Vector Laboratories RL-1062, RRID:AB_2336769), CD11c with Alexa Fluor 488 or 647 conjugated anti-CD11c (1:100, Biolegend 117311, RRID:AB_389306; 1:100, Biolegend 117312; RRID:AB_389328), ZO1 with mouse anti-ZO1 (1/100; ThermoFisher, 33–9100, RRID:AB_2533147), nuclei with DAPI (1:5,000; ThermoFischer 62247). Secondary antibodies were Alexa Fluor 488 or 555 goat anti-mouse or anti-rabbit IgG (1:500, Invitrogen).

## RNA isolation and quantitative real-time PCR

Small intestine (duodenum to ileum) was homogenized in 7 mL tubes containing a mix of 1.4 and 2.8 mm glass beads and 2 mL Trizol (ThermoFisher), respectively using the Precellys system (Bertin Technologies). RNA extraction from organs and cultured cells, RNA clean-up and DNase treatment were performed using the Nucleospin RNA kit (Macherey-Nagel) according to the manufacturer's instructions.

RNA was quantified using the NanoDrop ND-100 (ThermoFisher) and cDNA synthesis was performed from 1 µg of RNA using oligo-dT (ThermoFisher) and SuperScript IV RT (ThermoFisher). RT-qPCR was performed using Power SYBR Green PCR system (ThermoFisher) and specific primers listed in *Supplementary file 1* on the LightCycler 480 Real-Time PCR System (Roche). Relative quantification of gene expression was performed using the comparative $2^{-\Delta\Delta Ct}$ method. Results were normalized using *Actin* as the housekeeping gene.

## Flow cytometry analysis

Cells from MLN were mechanically dissociated with a 40 µm filter. Freshly isolated cells were stained with Live/Dead Fixable Aqua Dead Cell Stain Kit for 405 nm excitation (Thermofisher L34957) 20 min at 4°C, incubated with anti-CD16/32 (Biolegend 101301, RRID:AB_312800) in FCS (2% in PBS) 20 min at 4°C, and stained with the following antibodies: Brilliant Violet 785 anti-CD11c (Biolegend 117335, RRID:AB_11219204), PE/Cy5 anti-F4/80 (Biolegend 123111, RRID:AB_893494), Brilliant Violet 711 anti-CD19 (Biolegend 115555, RRID:AB_2565970), APC anti-CD45 (Biolegend 103111, RRID: AB_312976), FITC anti-mouse I-A/I-E (anti-MHC-II) (Biolegend 107605, RRID:AB_313320). Antibodies were used according to the manufacturer's instructions. For each sample, 50,000 to 100,000 viable cells were analyzed using a Fortessa flow cytometer (BD biosciences) and the proportion of CD45+, CD19-, F4/80-, CD11c+, CMH2+ cells among Live/Dead negative cells was calculated using the Kaluza software (Beckman Coulter, RRID:SCR_016182).

## Adherence, invasion and transcytosis assays

Human Caco-2 cells were washed twice in PBS and infected by their apical side at a multiplicity of infection of 10 (adherence and invasion assays) and 100 (transcytosis assay). Cells were incubated with bacteria 2 hr at 37°C in a 10% $CO_2$ atmosphere, washed 4–5 times with PBS and cell lysates recovered for quantification of adherent bacteria. For quantification of invasive bacteria, cells were incubated 2 hr with the bacteria, washed four times with PBS and incubated at 37°C in a 10% $CO_2$ atmosphere for two additional hours in fresh medium containing 200 mg/L gentamicin before washing and lysis. For transcytosis assay, cells were incubated with bacteria 2 hr at 37°C in a 10% $CO_2$ atmosphere, the medium of the lower compartment of the cell culture chamber was sampled for CFU counts and cells were fixed for staining or lysed for gene and protein expression analysis. Adherence, invasion and transcytosis were expressed as a percentage of the bacterial inoculum.

## Western-blotting

Cells were lysed in 300 µL of radio immune precipitation assay (RIPA) buffer and the lysate heated in Laemmli buffer for 5 min at 90°C before migration in acrylamide SDS-PAGE. Proteins were transferred onto PVDF membranes, incubated with the anti-GP2 antibody (1:300, abcam ab105503, RRID:AB_10859272) overnight at 4°C in PBS/BSA 5%, washed in PBS/Tween 0.1%, incubated with a peroxidase-labeled secondary antibody (1:10,000) for 1 hr and revealed with ECL chemiluminescence reagent (ThermoFisher). Acquisitions and image quantifications were performed with a Fusion Fx Imager (Vilber). Experiments were performed three times in triplicate.

## Reagents

Pierce LDH Cytotoxicity Assay Kit was from Thermofisher. Other reagents, estradiol (E2758), progesterone (P8783), sesame oil (S3547), $CaCO_3$ (C4830), Lucifer Yellow dipotassium salt (L0144), gentamicin (G1397) were from Sigma-Aldrich.

Cytokines dosage was performed by Luminex technology (Meso Scale Discovery).

Estradiol and progesterone dosage were performed by electrochemiluminescence on the Cobas 8000 analyzer (Roche).

Mouse serum anti-GBS was generated as follows. GBS (strain NEM316) were grown in Todd Hewitt broth to mid-exponential phase ($OD_{600nm}$ = 0.5), harvested by centrifugation (10 min at 4,000 g), and $3.10^9$ CFU were suspended and inactivated in 1 mL PFA (4% in PBS) for 30 min. Inactivated bacteria were washed three times in PBS and suspended in 1 mL PBS. Two SPF mice (8-week-old females BALB/c, Janvier, France) were immunized by subcutaneous injection of 50 µL inactivated bacteria + 50 µL complete Freund's adjuvant at day 1, followed by subcutaneous injection of 50 µL inactivated bacteria + 50 µL incomplete Freund's adjuvant at day 15 and day 30. Mice were sacrificed at day 35 and the serum was collected and stored at −20°C.

## Image acquisition and analysis

Images from intestinal ligated loops (2–3 per animal) were randomly acquired using the Leica DMI6000 confocal microscope. Areas of interest including M cells were acquired. Ten images (three experiments in triplicate) of cultured Caco-2 cells in each condition were randomly acquired using

the Leica DMI6000 confocal microscope or the wide field Leica 500A confocal microscope for quantitative imaging.

M cells and GBS were quantified using immunofluorescence with the ImageJ software (RRID: SCR_003070). For each image, we applied a protocol to measure the total surface of M cells, enterocytes and GBS, as well as the surface of GBS associated with M cells or enterocytes. Collected measurement values were exported to Microsoft Excel. The relative frequency of GBS associated with M cells was determined as a percentage of the total GBS surface from each individual image.

## Statistical analysis

Data were analyzed using GraphPad Prism 5.0 software. Multiple-group comparisons were performed by Kruskal-Wallis test followed by Dunn's multiple comparisons test or non-parametric two-way ANOVA followed by Tukey's multiple comparisons test. Single-group comparisons were performed by Mann-Whitney test. A p value < 0.05 was considered significant.

Statistical details of the experiments can be found in the figure legends.

## Study approval

Work on animals was performed in compliance with French and European regulations on care and protection of laboratory animals (EC Directive 2010/63, French Law 2013–118, February 6, 2013). All experiments were approved by the Ethics Committee. Animals were not involved in any previous procedure.

## Acknowledgements

This work was supported by the FRM, grant DBF20160635740. The funders had no role in study design, data collection and analysis, decision to publish, or preparation of the manuscript.

We thank Matthieu Benard from the animal facility of the Institut Cochin for his help with animal experiments, Karine Bailly and Patrice Vallin from the Cytometry and Immunobiology facility of the Institut Cochin for help in flow cytometry experiments, Antonin Weckel for advices in quantitative imaging, and Céline Plainvert for assistance. We are also grateful to Shaynoor Dramsi and Patrick Trieu-Cuot for critical reading of the manuscript.

## Additional information

### Funding

| Funder | Grant reference number | Author |
|---|---|---|
| Fondation pour la Recherche Médicale | DBF20160635740 | Constantin Hays<br>Gérald Touak<br>Claire Poyart<br>Asmaa Tazi |

This work was supported by the FRM, grant DBF20160635740. The funders had no role in study design, data collection and analysis, decision to publish, or preparation of the manuscript.

### Author contributions

Constantin Hays, Conceptualization, Resources, Investigation, Methodology, Writing—original draft, Writing—review and editing; Gérald Touak, Resources, Investigation; Abdelouhab Bouaboud, Resources, Data acquisition, Analysis and interpretation of data; Agnès Fouet, Julie Guignot, Methodology, Writing—review and editing; Claire Poyart, Conceptualization, Supervision, Methodology, Writing—review and editing; Asmaa Tazi, Conceptualization, Resources, Supervision, Investigation, Methodology, Writing—original draft, Writing—review and editing

### Author ORCIDs

Asmaa Tazi https://orcid.org/0000-0001-9531-9177

## Ethics

Animal experimentation: Work on animals was performed in compliance with French and European regulations on care and protection of laboratory animals (EC Directive 2010/63, French Law 2013-118, February 6, 2013). All experiments were approved by the Ethics Committee of the Paris Descartes University (Permit numbers APAFIS#390 and APAFIS#17106). Animals were not involved in any previous procedure.

## Decision letter and Author response

Decision letter https://doi.org/10.7554/eLife.48772.013
Author response https://doi.org/10.7554/eLife.48772.014

## Additional files

### Supplementary files

• Supplementary file 1. Primers used for RT-qPCR.
DOI: https://doi.org/10.7554/eLife.48772.010

• Transparent reporting form DOI: https://doi.org/10.7554/eLife.48772.011

### Data availability

All data generated or analysed during this study are included in the manuscript.

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
