## [Decision Letter]

**Acceptance summary:**

This study is important as it constitutes a new key step in the elucidation of the mechanisms of Group B *Streptococcus* Late Onset infection after the first week of life. The results contained in the manuscript show that under conditions representing hormone levels after 7 days post birth, infection with the GBS leads to meningitis as the bacteria are better able to cross the intestinal barrier compared to conditions representing hormone levels at birth. The success of strains of GBS CC17 clonal complex is explained by their expression of the surface protein *srr2* that allows invasion of the intestinal barrier. The results presented here are of general interest to *eLife* readers as they reveal that a commensal opportunistic pathogen can take advantage of the effect of hormonal levels on intestinal cell differentiation to invade the host. This work paves the way for research on other pathogens responsible for Late Onset infections.

**Decision letter after peer review:**

Thank you for submitting your article "Perinatal hormones favor CC17 Group B *Streptococcus* intestinal translocation through M cells and hypervirulence" for consideration by *eLife*. Your article has been reviewed by four peer reviewers, and the evaluation has been overseen by a Reviewing Editor and Wendy Garrett as the Senior Editor. The reviewers have opted to remain anonymous.

The reviewers have discussed the reviews with one another and the Reviewing Editor has drafted this decision to help you prepare a revised submission.

All the reviewers agree that this is quite an interesting study showing the effect of pregnancy hormones (estradiol and progesterone) on the late onset of neonatal GBS sepsis. Using a mouse model of hormonal exposure and oral bacterial challenge, you showed that conditions representing hormone levels after 7 days post birth favor meningitis following infection with GBS CC17 rather than conditions representing hormone levels at birth, because of a better translocation across the intestinal barrier through the Peyer's patches. Your work also reveals an interesting effect of the Srr2 protein in modulating the interaction of GBS CC17 BM110 with intestinal cells *in vitro*. The work is important and would be of interest to *eLife* readers, pending revision of a few key points.

Essential revisions:

- Without controls, the presumed role of high neonatal hormone concentrations remains unclear. No control for the hormone treatment (just vehicle) is provided. Without a 'no hormone' control it is difficult to determine if C_7_ levels of E2 and P4 promote infection or if higher levels of E2 and P4 at C_0_ supress infection. Please add this control, a dose-response experiment and provide growth curves of GBS in presence of hormones.

- Is the effect of Srr2 specific for the enhanced virulence of CC17 and is it common to other CC17 isolates? Is the effect of Srr specific for the enhanced virulence of CC17 at E2-P4C_7_, since tissue and organ invasion are also reduced in E2-P4 C_0_?

- Since Srr2 is responsible for adhesion to M cells in the *in vitro* model and for GBS invasion in the mouse model, could the authors demonstrate that Srr2 is actually responsible for the translocation across the M cells? The authors should use their *in vitro* model used to demonstrate translocation in Figure 3 to demonstrate this using the Srr2 wild type and isogenic mutant.

- Perhaps non parametric tests might be more appropriate than t tests?

---

## [Author Response]

Essential revisions:- Without controls, the presumed role of high neonatal hormone concentrations remains unclear. No control for the hormone treatment (just vehicle) is provided. Without a 'no hormone' control it is difficult to determine if C_7_ levels of E2 and P4 promote infection or if higher levels of E2 and P4 at C_0_ supress infection. Please add this control.

The “no hormone” control has been added in the experiments showing the impact of hormonal levels on bacterial dissemination and intestinal translocation in mice (Figure 1 and 2) as well as in the experiments analyzing the impact of hormones on bacterial translocation through M cells *in vitro* (Figure 3 and 4).

This control was actually already performed before submitting the first version of the manuscript in almost all experiments (those shown in Figure 1 and 3 but not those in Figure 2 and 4). Given the fact that it showed no significant difference with the E2-P4 C_0_ condition, we thought not to show it. As requested, this control is now provided in the revised version. It clearly shows no significant difference with the highest levels of E2 and P4 at C_0_, indicating that this latter condition does not obviously suppress the infection, which is now specified in the main text of the manuscript wherever necessary (“and control”).

Data shown on Figure 2 are the results of new experiments. We confirm that E2-P4 C_7_ hormonal concentrations increase the number of intratissular bacteria in the Peyer’s patches but we did not find, as previously written, that intratissular bacterial counts in the Peyer’s patches were increased compared to intratissular bacterial counts in the intestine. Therefore, the sentence “the number of intratissular bacteria was ~20 times higher in the Peyer’s patches than in the rest of the intestine” has been suppressed in the revised version of the manuscript (subsection “CC17 GBS crosses the intestinal barrier through Peyer’s patches in a process favored by E2-P4 C_7_ concentrations”, first paragraph). We think that this difference is due to mice susceptibility to infection which can greatly varies from a litter to another and which was increased in our latest experiments. Nevertheless, it does not change the main conclusion of the corresponding paragraph which remains that E2-P4 C_7_ hormonal condition is associated with “an increased crossing of the intestinal barrier by CC17 GBS through the Peyer’s patches”.

Please add a dose-response experiment.

Concerning the request to provide a dose-response experiment, we disagreed for both scientific and ethical reasons listed below.

i) The “control” condition in mice corresponds to mice which were administered the vehicle (sesame oil) alone instead of vehicle + hormones. These mice do actually synthetize physiological basal levels of E2 and P4 which were quantified in their serum. Circulating levels of E2 and P4 in control mice are approx. 100 times and 1,000 times lower than those of E2-P4 C_7_ and E2-P4 C_0_ mice, respectively. Thus, the 3 groups of mice already represent 3 different doses of hormones.

ii) Adding a fourth group of mice with levels of E2 and P4 10 times lower than E2-P4 C_7_ mice would have been difficult to interpret. Indeed, the effect of hormones and especially of E2 is biphasic and not strictly dose-dependent. For instance, high levels of E2 are anti-inflammatory whereas low levels are pro-inflammatory (Straub, 2007). Therefore, mice impregnated with E2-P4 levels 10 times lower than C_7_ mice could be either more susceptible than C_7_ mice, or conversely similar to control mice. Besides, given the important inter-individual variability of hormonal concentrations during infancy, these concentrations do not correspond to a precise period of child development and can be found between 2 and 5 months of age (Kuijper et al., 2013; Schaadt, Brain Lang, 2015).

iii) To statistically analyze the effect of this additional hormonal condition, we should have repeated all the animal experiments already performed with 3 groups of mice with the 4 groups of mice, which would have importantly extended the number of animals used in this study without providing real insights to the main question raised in the paper, i.e. whether and how hormonal levels found at LOD onset, corresponding to the E2-P4 C_7_ condition, could specifically favor CC17 GBS infection in comparison to hormonal levels found at birth.

In conclusion, for all these reasons, we decided not to add this experiment and we hope that you will agree with these considerations.

Please provide growth curves of GBS in presence of hormones.

Growth curves of GBS in the different conditions are provided and show no difference. This result has been added as a new figure (Figure 1—figure supplement 1) and is commented in the subsection “E2-P4 circulating levels modulate the severity of CC17 GBS meningitis in mice”.

- Is the effect of Srr2 specific for the enhanced virulence of CC17 and is it common to other CC17 isolates?

Srr2 belongs to a large family of serine-rich repeat glycoproteins widespread in streptococci and staphylococci. Molecular epidemiological data using either gene sequencing (Brochet et al., 2006) or whole genome sequencing (Da Cunha et al., 2014) of more than 200 GBS strains revealed that Srr2 is always and almost exclusively present in CC17 strains, whereas the non-CC17 isolates express another serine-rich repeat protein called Srr1. Despite structural similarities, Srr1 and Srr2 only display 32% of sequence identity at the amino-acid level (Six et al., 2015). Therefore, the impact of Srr2 on CC17 virulence is considered specific and common to other CC17 isolates. These epidemiological data regarding Srr2 distribution in GBS strains are more clearly explained in the Results subsection “The surface protein Srr2 contributes to CC17 GBS interaction with M cells and to the severity of meningitis in E2-P4 C_7_ condition” and in the first paragraph of the Discussion section.

Is the effect of Srr specific for the enhanced virulence of CC17 at E2-P4 C_7_, since tissue and organ invasion are also reduced in E2-P4 C_0_?

The fact that the virulence of the *∆srr2* mutant is not enhanced in E2-P4 C_7_ mice suggests that the increased virulence of the WT CC17 in this condition is Srr2 dependent. We acknowledge that we cannot exclude the possibility that the intestinal translocation abilities of the *∆srr2* mutant could be so altered that we could not observe an increased translocation of the mutant in E2-P4 C_7_ condition. However, this possibility seems very unlikely given the fact that the capacity of the *∆srr2* mutant to cross the intestine is not completely abolished, with median intratissular bacterial counts in the Peyer’s patches of approx. 500 CFU/g of tissue. This point is discussed in the first paragraph of the Discussion section, and we tempered our initial conclusion that Srr2 contributes to the enhanced virulence in E2-P4 C_7_ condition accordingly, replacing “demonstrating that Srr2 participates to the increased virulence of CC17 GBS in E2-P4 C_7_ hormonal condition” by “suggesting that Srr2.

*- Since Srr2 is responsible for adhesion to M cells in the* in vitro *model and for GBS invasion in the mouse model, could the authors demonstrate that Srr2 is actually responsible for the translocation across the M cells? The authors should use their* in vitro *model used to demonstrate translocation in Figure 3 to demonstrate this using the Srr2 wild type and isogenic mutant.*

We thank the reviewers for requesting this experiment which was indeed missing. The capacity of the *∆srr2* mutant to cross the monolayer composed of enterocytes and M cells has been assessed and the results showing that Srr2 actually contributes to CC17 transcytosis have been added to Figure 5 (shown as Figure 5C) and commented in the first paragraph of the subsection “The surface protein Srr2 contributes to CC17 GBS interaction with M cells and to the severity of meningitis in E2-P4 C_7_ condition”.

- Perhaps non parametric tests might be more appropriate than t tests?

All the statistical tests that were performed were actually already non-parametric tests but we omitted to make it clear. We apologize for this misunderstanding and the mistake is now repaired, as specified in the Materials and methods subsection “Statistical analysis” and in the legends of the figures.